# Prognostic Significance of Key Molecular Markers in Thyroid Cancer: A Systematic Literature Review and Meta-Analysis

**DOI:** 10.3390/cancers17060939

**Published:** 2025-03-10

**Authors:** Linh T. T. Nguyen, Emma K. Thompson, Nazim Bhimani, Minh C. Duong, Huy G. Nguyen, Martyn Bullock, Matti L. Gild, Anthony Glover

**Affiliations:** 1Kolling Institute, Northern Sydney Local Health District and Sydney Medical School, Faculty of Medicine and Health, University of Sydney, Sydney, NSW 2050, Australia; thng0079@uni.sydney.edu.au (L.T.T.N.); martyn.bullock@sydney.edu.au (M.B.); matti.gild@sydney.edu.au (M.L.G.); 2Department of Endocrinology, The 108 Military Central Hospital, Hanoi 100000, Vietnam; 3The Kinghorn Cancer Centre, Garvan Institute of Medical Research, St. Vincent’s Clinical School, Faculty of Medicine, University of New South Wales, Darlinghurst, NSW 2010, Australia; e.thompson.1@unsw.edu.au (E.K.T.); nazim.bhimani@sydney.edu.au (N.B.); 4Specialty of Surgery, Sydney Medical School, Faculty of Medicine and Health, The University of Sydney, Camperdown, NSW 2050, Australia; 5School of Population Health, University of New South Wales, Sydney, NSW 2033, Australia; minh.duong@unsw.edu.au; 6School of Biomedical Engineering, University of Technology Sydney, Sydney, NSW 2007, Australia; gia.h.nguyen@student.uts.edu.au; 7Department of Endocrinology, Royal North Shore Hospital, Northern Sydney Local Health District, Sydney, NSW 2065, Australia

**Keywords:** thyroid cancer, mutations, *TERT* promoter, *TP53*, PI3K pathway, *BRAF*, survival

## Abstract

This study explores the effect of genetic mutations identified via Next-Generation Sequencing (NGS) on overall survival (OS) and disease-free survival (DFS) in thyroid cancer patients. A meta-analysis of nine studies encompassing 1075 patients, sourced from MEDLINE, Scopus, and EMBASE, reveals significant findings. *TERT* promoter mutations correlate with poorer OS (pooled HR = 1.90, 95% CI: 1.17–3.09) and DFS (pooled HR = 2.76, 95% CI: 1.86–4.10). Additionally, *TP53* and PI3K pathway mutations are associated with shorter OS, with pooled HR = 2.87, 95% CI: 1.44–5.86 and HR = 2.17, 95% CI: 1.05–4.15, respectively. *BRAF* mutations, however, show no significant link to OS or DFS. These findings suggest that *TERT*, *TP53*, and PI3K pathway mutations are valuable for risk stratification, while *BRAF* lacks predictive utility. Further research is necessary to clarify the relationship between *TP53* and PI3K pathway mutations and DFS outcomes.

## 1. Introduction

Thyroid cancer is the most common malignancy of the endocrine system and ranked ninth in global cancer incidence in 2020, underscoring its widespread prevalence and clinical significance [1]. Most thyroid cancers are well-differentiated, exhibit favorable prognoses, and have high survival rates. However, aggressive subtypes, such as medullary and anaplastic thyroid cancers, pose significant challenges due to higher recurrence rates and resistance to I-131 therapy, leading to poor clinical outcomes [2].

The pathogenesis of thyroid cancer often begins with a single genetic alteration in either the mitogen-activated protein kinase (MAPK) pathway or phosphatidylinositol 3-kinase (PI3K) pathway, leading to uncontrolled cell growth, proliferation, and apoptosis. Progression to malignancy involves the accumulation of additional mutations that activate oncogenes or inactivate tumor suppressor genes, driving aggressive behavior. Advanced stages of thyroid cancer may metastasize to lymph nodes, lungs, or bones and can develop resistance to radioiodine therapy, rendering poorer survival. This may require alternative treatments, including targeted therapies and chemotherapy [2].

Molecular testing has become the standard procedure in managing thyroid cancer in the United States, significantly enhancing understanding of its molecular drivers and guiding targeted therapeutic strategies [3]. *BRAF^V600E^*, hereafter referred to as *BRAF*, is the most prevalent and extensively studied; however, its prognostic value in predicting survival, recurrence, or progression remains inconclusive [2]. Instead, it is the role of secondary mutations that has been explored in improving prognostic accuracy. *TERT* promoter mutations at positions c-124 C > T (C228T) and c-146 C > T (C250T), hereafter referred to as *TERT*, have emerged as the most reliable markers of poor prognosis, independently associated with distant metastasis and increased disease-specific mortality [2,4,5]. Additionally, PI3K/AKT pathway and *TP53* mutations are linked to worse outcomes, although their roles as independent predictors are less well-defined [6,7,8]. Despite significant advances, a clear consensus on the prognostic utility of secondary mutations beyond *BRAF* in thyroid cancer is lacking, leaving clinicians with limited guidance for integrating these markers into risk stratification frameworks.

This systematic review and meta-analysis aims to evaluate the prognostic significance of *BRAF*, *TERT*, *TP53*, and PI3K pathway mutations in follicular-derived thyroid cancers to provide a more comprehensive understanding of their impact on clinical outcomes.

## 2. Materials and Methods

### 2.1. Literature Search

A comprehensive search was conducted across three electronic databases—PUBMED, Scopus, and EMBASE—for articles on thyroid cancer patients of all ages from inception to April 2024. The search strategy employed a combination of keywords and subject headings tailored to each database’s indexing system, with detailed methodologies outlined in Appendix A. Additional studies were identified by reviewing citations from included articles and relevant reviews. This systematic review protocol was registered on PROSPERO (registration date: 13 August 2024) and complies with PRISMA-P guidelines. The protocol is accessible at https://www.crd.york.ac.uk/PROSPERO/view/CRD42024579106, accessed on 24 August 2024. 

### 2.2. Selection Criteria and Abstract Screening

All search results from electronic databases were imported into Covidence for duplicate removal. Two reviewers independently screened titles and abstracts. Studies were included if they employed NGS testing to detect genetic alterations and examined associations between *TERT*, *TP53*, or PI3K pathway mutations (with or without *BRAF*) and outcomes such as overall survival (OS), disease-free survival (DFS), or recurrence (RC). Eligible studies involved patients of all ages with follicular-derived thyroid cancer, including differentiated (DTC), poorly differentiated (PDTC), and anaplastic (ATC) subtypes. Studies on DTC have a follow-up duration of at least 1 year to allow sufficient time for events to occur, while the follow-up period in ATCis limited to six months, reflecting the average survival time of these cohorts, which typically ranges from 4 to 10 months [9,10,11].

Exclusions applied to studies focusing on non-DTC/ATC subtypes, those solely investigating *BRAF* without other mutations, studies lacking follow-up, case reports, reviews, posters, conference papers, theses, books, or duplicates. For multiple publications on the same cohort, the study with the most relevant data was included. Reasons for exclusions were documented, and disagreements between the two reviewers were resolved through discussion. Authors were contacted for missing or unpublished data; the study was excluded if the author did not respond.

### 2.3. Outcomes Definition

OS was the primary outcome, defined as the time from diagnosis to death or the last follow-up. In studies of DTC, DFS was defined as the time from the completion of initial treatment (e.g., surgery, radioactive iodine) to the first occurrence of clinical or radiological recurrence or biochemical recurrence indicated by elevated thyroglobulin (Tg) levels, with specific Tg thresholds as defined by individual studies. In studies of ATC, DFS was defined as the time from the completion of initial treatment (e.g., surgery, radiation, chemotherapy) to the first occurrence of clinical or radiological recurrence [3,12].

### 2.4. Full-Text Screening and Data Extraction

Two reviewers independently screened the full texts of all relevant studies. Data were extracted into a predefined form, capturing information on the study title, authors, publication year, design, country, ethnicity, institution, cancer type, participant numbers, age, sex, follow-up periods, mutation detection methods, molecular tests (*BRAF*, *TERT*, PI3K), and hazard ratio (HR) data with 95% confidence intervals (CIs) for OS, DFS, and RC, along with any adjusted variables where available. HR and 95% CI values were either directly obtained from the full texts or estimated using Kaplan–Meier curves (KMC) following Herbert et al.’s methods [13].

(1)If both adjusted and unadjusted HRs were provided, the unadjusted HR was used.(2)The HR was estimated using the number of patients and events in each group and the log-rank *p*-value.(3)The HR was estimated from the KMC and data on the number of patients at risk. Individual patient data were reconstructed using the Guyot algorithm, and KMC points were extracted using the DigitizeIt software (https://www.digitizeit.xyz, version 2.6).(4)HRs estimated from the KMC without data on patients at risk were calculated using a single time point or median survival time and the log-rank *p*-value.

The survival data in Zehir et al. and Nguyen et al. papers were extracted from the cBio-Portal for Cancer Genomics (https://www.cbioportal.org/).

### 2.5. Quality Assessment

The quality of studies included in the meta-analysis was assessed using the Newcastle-Ottawa Scale (NOS), which employs a star-based rating system. Studies were evaluated on three criteria: up to four stars for group selection, two stars for comparability, and three stars for outcomes, with a maximum of nine stars. Studies scoring at least six stars were considered moderate to high quality, while those with fewer than six were deemed low quality. Stars were awarded based on a predefined checklist, which included key confounding variables such as age and sex. In the outcomes section, a star was given for studies with a median follow-up of over five years for DTC or six months for ATC and those with follow-up rates above 80% or a clear description of participants lost to follow-up.

### 2.6. Analysis

Statistical analysis was performed using R version 4.2.1. The pooled HR of OS, DFS, and RC was calculated using a random-effects model with inverse variance weighting. An HR greater than 1 indicated a worse prognosis for patients with mutations. Forest plots were generated to visualize the associations between the mutations and outcomes. Heterogeneity was assessed using the I^2^ statistic, with values <25% indicating low heterogeneity and >50% indicating high heterogeneity. Subgroup and sensitivity analyses explored potential sources of heterogeneity. Publication bias was evaluated using Egger’s regression test and funnel plots, with a *p*-value < 0.05 considered indicative of significant bias.

### 2.7. Assessing the Quality of Evidence

We followed the GRADE approach described in the Cochrane Handbook for Systematic Reviews of Interventions [14]. Two reviewers independently assessed the quality of the evidence, and any discrepancies were resolved through discussion or by consulting a third reviewer. The assessment involved evaluating the evidence across several domains, including risk of bias, inconsistency, indirectness, imprecision, and publication bias. Each domain was scored, and the overall quality of evidence for each outcome was rated as high, moderate, low, or very low, starting from an initial rating of “high” for randomized trials and “low” for observational studies, then adjusted based on these factors.

This systematic review and meta-analysis were conducted and reported in accordance with the Preferred Reporting Items for Systematic Reviews and Meta-Analyses (PRISMA) guidelines [15].

## 3. Results

### 3.1. Study Selection

A total of 5893 records were identified through database searches in PubMed (*n* = 1884), Embase (*n* = 2167), and Scopus (*n* = 1842), with an additional 20 records identified from other sources. After duplicate removal and screening, 132 studies were assessed for eligibility. Nine studies met the inclusion criteria and were included in the systematic review and meta-analysis (Figure 1).

### 3.2. Characteristics of Included Studies

The nine included studies encompassed 37,129 patients, with sample sizes ranging from 40 to 25,775. Among these, 1075 patients with DTC/PDTC or ATC who had NGS results were included in the meta-analysis. The mutations analyzed included *BRAF*, *TERT, TP53*, and PI3K pathway alterations, with OS and DFS reported in most studies. None of the studies provided data on RC outcomes. Detailed characteristics of the studies are summarized in Table 1.

### 3.3. Prevalence of Mutations

#### 3.3.1. Differentiated and Poorly Differentiated Thyroid Cancer

The prevalence of genetic mutations in DTC and PDTC is summarized in Table 2. *BRAF* mutations were the most common, with a pooled prevalence of 70.69%. *TERT* promoter mutations were detected at a pooled rate of 45.3%, while PI3K pathway and *TP53* mutations were less frequent, with pooled rates of 13% and 10.8%, respectively.

#### 3.3.2. Anaplastic Thyroid Cancer

The genetic profile of ATC shows a distinct pattern, with an increased prevalence of *TERT* promoter, *TP53*, and PI3K pathway mutations compared to DTC (Table 3). The pooled prevalence of *BRAF* mutations decreased to 45.8%, whereas *TERT* promoter mutations increased significantly to 53.2%. Alterations in the PI3K pathway were observed in 27.5% of cases, and *TP53* mutations were the most prevalent, at 57.1%.

#### 3.3.3. Co-Mutations

In DTC, the most frequent co-mutations occurred between *BRAF* and *TERT*, with a pooled prevalence of 35.9%. Comutations between *TERT* and PI3K pathway mutations were rare, occurring at 3.9% (Table 2). In ATC, comutations between *TERT* and *TP53* were the most common, with a pooled prevalence of 33.4%, while comutations involving *TERT* and PI3K were less frequent, at 12.2% (Table 3).

### 3.4. Risk of Bias Assessment

The quality of the included studies was evaluated using the NOS. Seven studies scored seven or above, indicating high quality (Table 4). Two studies scored below 7, reflecting moderate quality due to limitations in outcome assessment or comparability.

Publication Bias: A funnel plot analysis revealed asymmetry in the outcomes of PI3K pathway mutations, suggesting potential publication bias (Figure 2. This bias underscores the need for cautious interpretation of findings.

### 3.5. Survival Outcomes Associated with Mutations

#### 3.5.1. *BRAF* Mutations

*BRAF* mutations are not significantly associated with OS (pooled HR = 1.11; 95% CI: 0.66–1.88) or DFS (pooled HR = 1.23; 95% CI: 0.66–2.29). Interestingly, in DTC patients, *BRAF* mutations are linked to longer survival (HR = 0.65, 95% CI: 0.46–0.94), while in ATC patients, they are associated with shorter survival. In two studies examining DFS, *BRAF* mutations showed no significant impact (HR = 1.23, 95% CI: 0.66–2.29) (Figure 3 and Figure 4).

#### 3.5.2. *TERT* Promoter Mutations

*TERT* mutations were strongly associated with decreased overall survival (HR = 1.90, 95% CI: 1.17–4.47) (Figure 5). In both DTC and ATC groups, *TERT* mutations are associated with shorter survival times. *TERT* mutations also strongly correlate with disease progression, as indicated by shorter disease-free survival times compared to the negative group (HR = 2.76, 95% CI: 1.86–4.10) (Figure 6).

#### 3.5.3. PI3K Pathway Mutations

PI3K mutations are associated with poorer overall survival (OS), with a pooled HR of 2.17 (95% CI: 1.05–4.51) (Figure 7). These mutations impact both DTC and ATC groups, though the effect appears more pronounced in the DTC group (HR = 2.99) compared to the ATC group (HR = 1.95). However, neither subgroup achieved statistical significance.

#### 3.5.4. *TP53* Mutations

The presence of *TP53* mutations was significantly associated with poorer survival outcomes, reflected by a marked reduction in overall survival (OS) (HR = 2.87, 95% CI: 1.49–5.53) (Figure 8). However, in the ATC subgroup, the association did not reach statistical significance (HR = 2.79, 95% CI: 0.93–8.32).

Data regarding the relationship between PI3K, *TP53* mutations, and disease-free survival is still minimal and was only found in Wang’s report [20], where the HR for PI3K is 0.75 (95% CI: 0.31–1.83), and the HR for *TP53* is 4.5 (95% CI: 1.90–10.70).

### 3.6. Summary of Findings

The quality of evidence for the associations between genetic mutations and OS and DFS was evaluated using the GRADE approach. Our Summary of Findings (SoF) table (Table 5) provides a concise overview of these associations, along with the GRADE ratings and comments on the limitations and strengths of the evidence. The SoF table indicates that *TERT* promoter mutations are strongly associated with poorer OS and DFS, and *TP53* and PI3K pathway mutations are associated with poorer OS. The quality of evidence for these associations is generally low to moderate, primarily due to the observational nature of the included studies.

## 4. Discussion

Although the exact sequence can vary, thyroid cancer progression commonly involves a stepwise accumulation of genetic mutations. Early alterations often occur in the MAPK pathway (e.g., *BRAF*, RAS), while subsequent mutations in *TERT*, *TP53*, and the PI3K/AKT pathway are generally associated with more aggressive tumor behavior and dedifferentiation [2]. Genetic analysis has become a critical tool in the diagnosis and management of thyroid cancer. Advancements in genomic testing—especially NGS enable the concurrent detection of multiple mutations linked to tumor malignancy [3]. This systematic review and meta-analysis synthesize data from nine studies investigating the genetic profiles of DTC and ATC, focusing on NGS results and survival outcomes. Our analysis underscores the prevalence and clinical significance of mutations in *BRAF*, *TERT*, the PI3K pathway, and *TP53*, shedding light on their roles in thyroid cancer progression and prognosis.

*BRAF* mutations, particularly at codon V600, are commonly found in PTC, occurring in approximately 60% of cases. In our study, the frequency of *BRAF* mutations was 45.8% in the ATC group and 70.69% in the DTC group. The prognostic value of *BRAF* mutations remains debated, with studies showing mixed results. One retrospective study found that lymph node metastasis (LNM) and *BRAF* significantly heightened the mortality risk in TC. Compared to the absence of both, the adjusted HR for coexisting LNM and *BRAF* was exceptionally high (HR = 27.39; 95% CI: 5.15–145.80), indicating a strong synergistic effect [25]. Another study observed that *BRAF*V600E was significantly associated with increased mortality, with an HR of 3.53 (95% CI, 1.25–9.98). However, after adjusting for factors like LNM and extrathyroidal invasion, the association was no longer significant. Furthermore, multivariable analysis adjusting for tumor behavior may invalidate the association with *BRAF* and mortality, which underestimates its prognostic significance [26]. In our meta-analysis, *BRAF* mutations did not show significant prognostic value for thyroid cancer in terms of OS (HR = 0.65; 95% CI: 0.46–0.94) or DFS (HR = 1.23; 95% CI: 0.66–2.29). These findings align with the 2015 American Thyroid Association (ATA) guidelines, which recommend against using *BRAF*^V600E^ alone as a prognostic marker for low-risk thyroid carcinoma due to its low positive predictive value [3].

*TERT* promoter mutations are common in various cancers, including thyroid carcinomas, where they enhance the proliferative potential of *BRAF*—or *RAS*-driven clones [4,27,28]. Their prevalence increases with tumor aggressiveness: fewer than 10% among DTC cases, but the rate rises in high-grade PTCs and PDTCs, exceeding 70% in ATCs [4]. Consistent with previous research, our study found that the prevalence of *TERT* mutations ranges from 26.5% to 59.6% in DTCs and 36.8% to 74.5% in ATCs.

In this study, *TERT* promoter mutations were significantly associated with reduced OS and DFS. These findings align with previous research, which has identified a robust link between *TERT* mutations and poorer clinical outcomes, including distant metastasis, recurrence, and mortality [4,5,28,29]. Our results further confirm the prognostic value of *TERT* mutations in the overall population of papillary thyroid cancer patients, demonstrating a nearly 1.9-fold increase in the risk of mortality (HR = 1.90, 95% CI: 1.17–4.77). However, the prognostic impact of *TERT* mutations was less pronounced within specific subgroups. Specifically, in the DTC group, *TERT* mutations did not show a statistically significant association with prognosis, with an HR of 1.89 (95% CI: 0.75–4.77). Interestingly, our sensitivity analysis, which excluded the study by Inguinez et al. due to its low NOS score of 4*, revealed a statistically significant prognostic value for *TERT* mutations, with an HR of 2.86 (95% CI: 1.15–7.10) (Appendix A). The relatively weaker prognostic effect of *TERT* mutations ATC (HR = 1.90) compared to DTC (HR = 2.86) may be attributed to the high mutation rate in ATC, which could mask the specific contribution of *TERT* mutations to clinical outcomes.

PI3K pathway mutations, observed in both DTC and ATC, are less frequent than *BRAF* and *TERT* mutations but are consistently linked to survival outcomes, underscoring their role in tumor progression. Mutations in *PIK3CA*, *AKT1*, and *PTEN* have been associated with aggressive, radioactive iodine-refractory thyroid tumors and are typically late events in thyroid cancer pathogenesis. Experimental models have demonstrated their role in driving cancer progression [30,31]. In this study, PI3K mutations were associated with a twofold increased mortality risk (HR = 2.17, 95% CI: 1.05–4.51) across the overall thyroid cancer population, though subgroup analyses lacked statistical significance, likely due to limited sample sizes.

Although research on the prognostic value of PI3K pathway mutations in thyroid cancer is limited, its significance in other cancers is well-documented. A meta-analysis of 1929 breast cancer cases identified *PIK3CA* mutations as independent negative prognostic factors, with a hazard ratio of 1.67 (95% CI: 1.15–2.43; *p* = 0.007) and an associated reduction in median OS by 8.4 months [32]. In gastric cancer, high *PIK3CA* protein expression correlates with poor prognosis, though gene amplification or mutation showed no significant prognostic link [33]. Additionally, a meta-analysis on PI3K/AKT/mTOR inhibitors in advanced solid tumors revealed improved progression-free survival when these inhibitors were added to treatment regimens [34]. These findings highlight the potential of PI3K pathway mutations as prognostic biomarkers and therapeutic targets in cancer.

Mutations in *TP53* are implicated in various cancers, including thyroid cancer. *TP53* encodes the p53 tumor suppressor protein, which regulates cellular responses to DNA damage, cell cycle arrest, and apoptosis. *TP53* mutations often result in the loss of these protective mechanisms, allowing uncontrolled proliferation, genomic instability, and resistance to apoptosis [35,36]. These effects are amplified in aggressive thyroid cancers, promoting resistance to conventional therapies such as radioactive iodine and external beam radiation [37]. As a hallmark of genomic instability, *TP53* mutations were prominent in ATC (63.1%) and correlated with poor prognosis in OS (HR = 2.87, 95% CI: 1.44–5.86) and shorter DFS (HR = 4.5; 95% CI: 1.90–10.70) in Wang’s study [20]. This finding reaffirms the role of *TP53* in aggressive cancer behavior and indicates the potential of *TP53* in thyroid cancer prognosis.

The identification of these genetic alterations has profound clinical implications. Mutations in *TERT*, *TP53*, and the PI3K pathway have emerged as strong biomarkers, all strongly associated with poor survival outcomes. A deeper understanding of the prognostic value of these genes can also guide the selection of targeted therapies, such as PI3K/AKT/mTOR pathway inhibitors, offering promising treatment options. These findings emphasize the importance of integrating genetic analysis into clinical decision-making for thyroid cancer patients, enabling more personalized and effective treatment strategies.

Despite its strengths, this review has several limitations. As shown in the funnel plot in Figure 8, publication bias may have impacted the findings. The included studies exhibited variability in methodology and population demographics, which could affect the generalizability of the results. Furthermore, some mutations, such as those in the PI3K pathway and *TP53*, were studied in smaller cohorts, potentially reducing the statistical power to detect meaningful associations. Additionally, the included studies did not provide explicit information on recurrence, which limits our ability to comprehensively evaluate the association between mutations and recurrence patterns and their potential implications. The quality of evidence in our findings, based on cohort studies, is inherently limited due to their observational nature, which establishes associations rather than causation and introduces potential biases. Therefore, the evidence quality is generally low to moderate, which requires cautious interpretation and further research to confirm these associations.

Future research should explore the synergistic effects of concurrent mutations, such as *TERT* + *BRAF* and *TERT* + PI3K pathway mutations, on disease progression and treatment response. More extensive multicenter studies are needed to improve the reliability of findings, especially for rare mutations. Investigating the biological mechanisms of these mutations could uncover new therapeutic targets, while long-term follow-up studies are essential to confirm the prognostic value of these genetic markers and guide effective, targeted treatments.

## 5. Conclusions

This systematic review and meta-analysis offer a comprehensive overview of the key genetic alterations in differentiated and anaplastic thyroid cancer, highlighting the clinical significance of mutations in *BRAF*, *TERT*, *TP53,* and PI3K pathway genes. The findings underscore the importance of genetic profiling in understanding tumor behavior, improving prognosis, and advancing targeted therapies for aggressive thyroid cancers. *TERT* promoter mutations were reaffirmed as strong prognostic biomarkers for poor outcomes, while *TP53* and PI3K mutations show potential as mortality risk markers, warranting further research. In contrast, *BRAF* mutations were confirmed to lack significant prognostic value in current analyses.

## Figures and Tables

**Figure 1 cancers-17-00939-f001:**
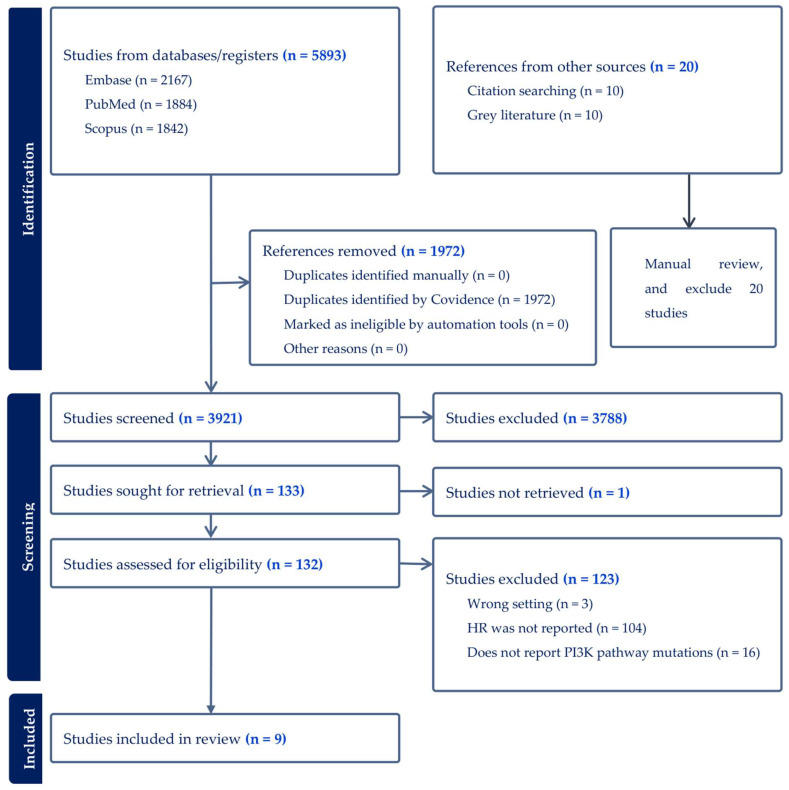
Selection and screening process for included studies from three databases.

**Figure 2 cancers-17-00939-f002:**
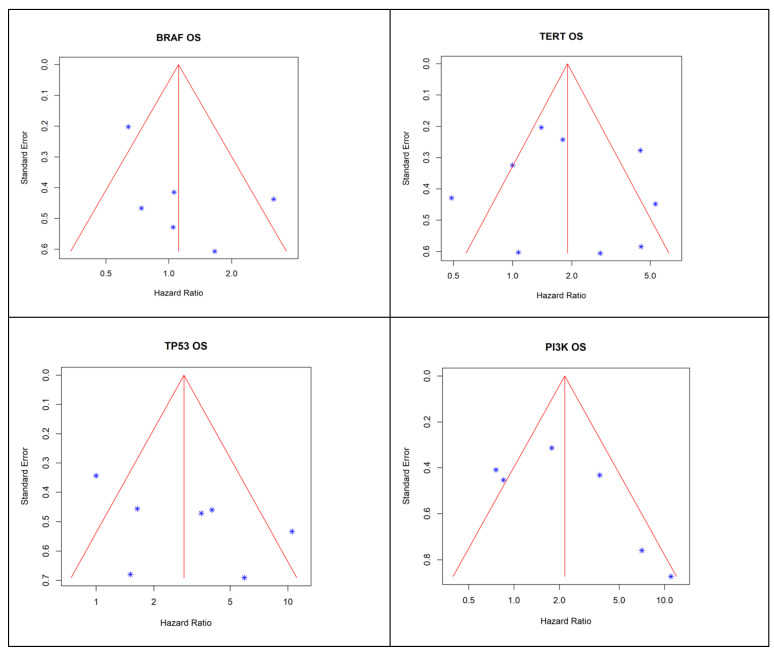
Funnel plots to assess publication bias. Each star (*) in the figure corresponds to a single study’s effect size plotted against the standard error.

**Figure 3 cancers-17-00939-f003:**
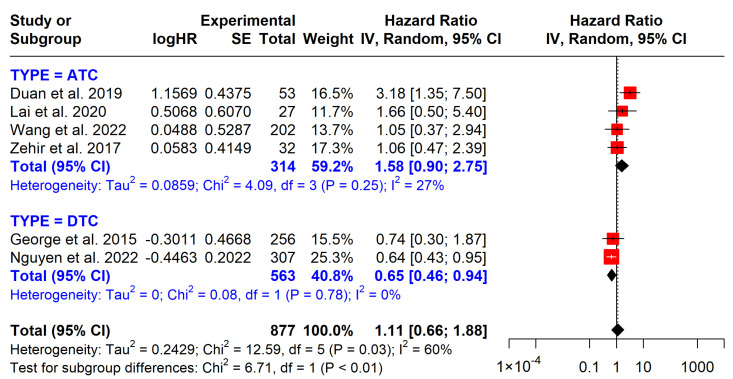
Impact of *BRAF* mutations on overall survival. CI = confidence interval; I2 = heterogeneity; df = degrees of freedom; SE: standard error. Each red square in the figure represents an effect size of a study and the area of the square represents the magnitude of a related study in the effect size. The lines on either side of the squares indicate the lower and upper limits in a 95% CI of the calculated effect sizes. The black rhombus at the bottom of the plot shows the calculated overall effect size.

**Figure 4 cancers-17-00939-f004:**
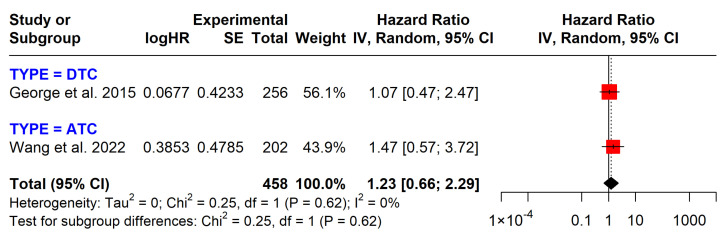
Impact of *BRAF* mutations on disease-free survival. CI = confidence interval; I2 = heterogeneity; df = degrees of freedom; SE: standard error. Each red square in the figure represents an effect size of a study and the area of the square represents the magnitude of a related study in the effect size. The lines on either side of the squares indicate the lower and upper limits in a 95% CI of the calculated effect sizes. The black rhombus at the bottom of the plot shows the calculated overall effect size.

**Figure 5 cancers-17-00939-f005:**
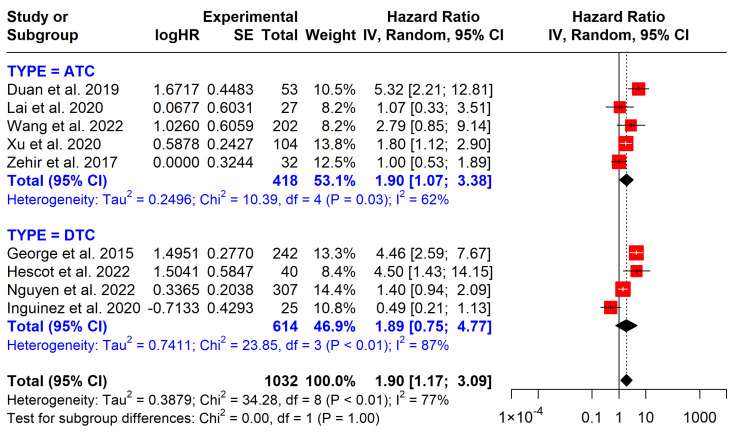
Impact of TERT mutations on overall survival. CI = confidence interval; I2 = heterogeneity; df = degrees of freedom; SE: standard error. Each red square in the figure represents an effect size of a study and the area of the square represents the magnitude of a related study in the effect size. The lines on either side of the squares indicate the lower and upper limits in a 95% CI of the calculated effect sizes. The black rhombus at the bottom of the plot shows the calculated overall effect size.

**Figure 6 cancers-17-00939-f006:**
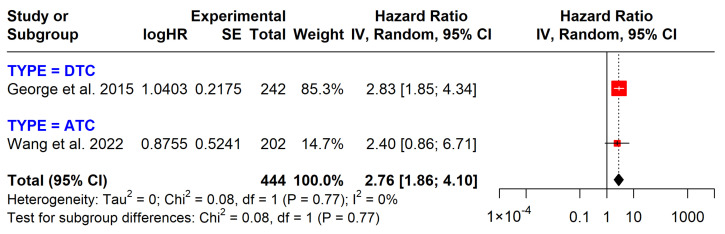
Impact of *TERT* mutations on disease-free survival. CI = confidence interval; I2 = heterogeneity; df = degrees of freedom; SE: standard error. Each red square in the figure represents an effect size of a study and the area of the square represents the magnitude of a related study in the effect size. The lines on either side of the squares indicate the lower and upper limits in a 95% CI of the calculated effect sizes. The black rhombus at the bottom of the plot shows the calculated overall effect size.

**Figure 7 cancers-17-00939-f007:**
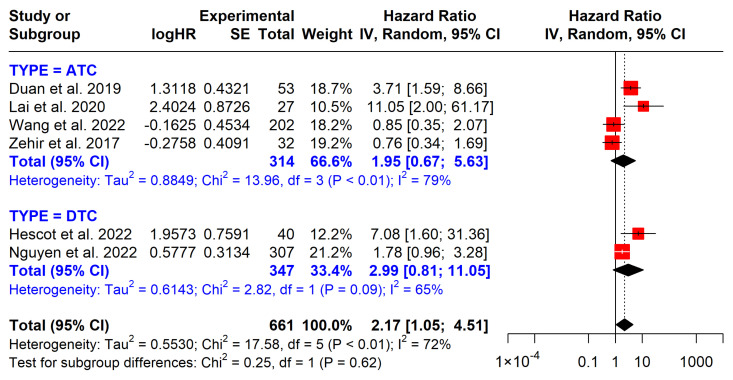
Impact of PI3K mutations on overall survival. CI = confidence interval; I2 = heterogeneity; df = degrees of freedom; SE: standard error. Each red square in the figure represents an effect size of a study and the area of the square represents the magnitude of a related study in the effect size. The lines on either side of the squares indicate the lower and upper limits in a 95% CI of the calculated effect sizes. The black rhombus at the bottom of the plot shows the calculated overall effect size.

**Figure 8 cancers-17-00939-f008:**
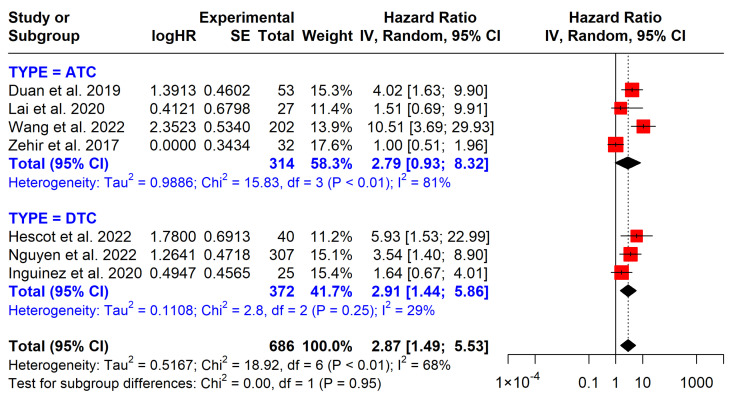
Impact of *TP53* mutations on overall survival. CI = confidence interval; I2 = heterogeneity; df = degrees of freedom; SE: standard error. Each red square in the figure represents an effect size of a study and the area of the square represents the magnitude of a related study in the effect size. The lines on either side of the squares indicate the lower and upper limits in a 95% CI of the calculated effect sizes. The black rhombus at the bottom of the plot shows the calculated overall effect size.

**Table 1 cancers-17-00939-t001:** Characteristics of nine included studies.

Author	Publication Year	Country	Cancer Type	N. of Patient	Female	Age	Observation Period	Mutation	Outcome	Genes Panel
Duan et al. [16]	2019	China	PDTC + ATC	66 (53 *)	29	NA	6 years	*BRAF*, *TERT*, PI3K, *TP53*	OS	Nextseq500, 18 genes
Hescot et al. [17]	2022	France	PDTC	40	24	59 (51–72) ^a^	57.5 months	*BRAF*, *TERT*, PI3K	OS, DFS	Thyroseq V3, 12 genes
Inguinez et al. [18]	2020	US	DTC/PDTC + ATC	55	26	55 (14) ^b^	30 years	*BRAF*, *TERT*.PI3K	OS	Foundation One, 315 genes
Lai et al. [19]	2020	Taiwan	ATC	39 (27 *)	20	75 (NA) ^a^	21 years	*BRAF*, PI3K, *RAS*, *TP53*	OS	Targeted NGS, 5 genes
Wang et al. [20]	2022	US	ATC	202	95	66 (32–92) ^a^	31 months	*BRAF*, *TERT*, PI3K, *TP53*	OS, DFS	Targeted NGS, 19 genes,
Xu et al. [21]	2020	US + AU	ATC	360 (102 *)	196	68 (29–99) ^a^	9 months	*BRAF*, *TERT*, PI3K, *TP53*	OS	MSK-IMPACT, 468 genes
Nguyen et al. [22]	2022	US	DTC	25775 (307 *)	154	NA	38 months	*BRAF*, *TERT*, PI3K	OS	MSK-IMPACT, 468 genes
Geoge et al. [23]	2015	US	PTC	256	108	45.3(NA) ^a^	112 months	*BRAF*, *TERT*,	OS, DFS	NGS, 46 genes
Zehir et al. [24]	2017	US	ATC	10336 (33 *)	14	NA	30 months	*BRAF*, *TERT*, PI3K, *TP53*	OS	MSK-IMPACT, 341 genes

* Number of sequenced patients; ^a^ median (Min-Max); ^b^ mean (SD); ATC: anaplastic thyroid cancer; DTC: differentiated thyroid cancer; PDTC: poorly differentiated thyroid cancer; OS: overall survival; DFS: disease-free survival; NA: no answer; *BRAF*: B-Raf proto-oncogene; PI3K/AKT: phosphatidylinositol-3-kinase/protein kinase B; *TERT*: telomerase reverse transcriptase; *TP53*: tumor protein 53.

**Table 2 cancers-17-00939-t002:** Prevalence of different mutations in DTC.

Author	Number	*BRAF* (*n*, %)	*TERT* (*n*, %)	PI3K (*n*, %)	*TP53* (*n*, %)	*RET* (*n*, %)	*RAS* (*n*, %)	*TERT* + *BRAF* (*n*, %)	*TERT* + *TP53* (*n*, %)	*TERT* + PI3K (*n*, %)
Duan et al. [16]	34	9 (26.5)	9 (26.5)	8 (23.5)	11 (32.4)	6 (17.6)	4 (11.8)	3 (8.8)	2 (5.9)	2 (5.9)
Hescot et al. [17]	40	2 (5.0)	17 (42.5)	7 (17.5)	9 (22.5)	-	17 (42.5)	2 (5.0)	2 (4.0)	4 (10.0)
Inguinez et al. [18]	25	10 (40.0)	14 (56.0)	5 (20.0)	9 (36.0)	1 (4.0)	3 (12.0)	9 (36.0)	3 (12.0)	0 (0.0)
Nguyen et al. [22]	307	212 (69.1)	183 (59.6)	33 (10.7)	15 (4.9)	22 (7.2)	36 (11.7)	147 (47.9)	12 (3.9)	18 (5.9)
Geoge et al. [23]	256	235 (91.8)	77 (30.1)	-	-	-	-	77 (30.1)	-	2 (0.8)
Pooled	662	468 (70.7)	300 (45.3)	53 (13)	44 (10.8)	29 (7.9)	60 (14.8)	238 (35.9)	19 (4.7)	26 (3.9)

*BRAF*, B-Raf proto-oncogene; PI3K/AKT: phosphatidylinositol-3-kinase/protein kinase B; RAS: rat sarcoma; RET: rearranged during transfection; *TERT*: telomerase reverse transcriptase; *TP53*: tumor protein 53; *n*: number of patients.

**Table 3 cancers-17-00939-t003:** Prevalence of different mutations in ATC.

Author	Number	*BRAF* (*n*, %)	*TERT* (*n*, %)	PI3K (*n*, %)	*TP53* (*n*, %)	*RET* (*n*, %)	*RAS* (*n*, %)	*TERT* + *BRAF* (*n*, %)	*TERT* + *TP53* (*n*, %)	*TERT* + PI3K (*n*, %)
Duan et al. [16]	19	14 (73.7)	14 (73.7)	11 (57.9)	15 (78.9)	0 (0)	7 (36.8)	8 (42.1)	10 (52.6)	6 (31.6)
Inguinez et al. [18]	19	6 (31.6)	13 (68.4)	6 (31.6)	14 (73.7)	2 (10.5)	4 (21.1)	4 (21.1)	11 (57.9)	3 (15.8)
Lai et al. [19]	27	7 (25.9)	22 (81.5)	4 (14.8)	19 (70.4)	-	11 (40.7)	6 (22.2)	15 (55.6)	3 (11.1)
Wang et al. [20]	202	84 (41.6)	46 (22.8)	31 (15.3)	119 (58.9)	-	45 (22.3)	20 (9.9)	31 (15.3)	8 (4.0)
Xu et al. [21]	102	46 (45.)	76 (74.5)	37 (36.3)	64 (62.7)	2 (2.0)	22 (21.6)	42 (41.2)	48 (47.1)	21 (20.6)
Zehir et al. [24]	32	14 (43.8)	5 (15.6)	14 (43.8)	22 (68.8)	0 (0)	7 (21.9)	14 (43.8)	19 (59.4)	8 (25.0)
Pooled	401	171 (42.6)	176 (43.8)	103 (25.7)	253 (63.1)	4 (2.3)	96 (23.9)	94 (23.4)	134 (33.4)	49 (12.2)

*BRAF*: B-Raf proto-oncogene; PI3K/AKT: phosphatidylinositol-3-kinase/protein kinase B; RAS: rat sarcoma; RET: rearranged during transfection; *TERT*: telomerase reverse transcriptase; *TP53*: tumor protein 53; *n*: number of patients.

**Table 4 cancers-17-00939-t004:** Newcastle-Ottawa Quality Assessment for cohort studies.

Author	Selection	Comparability	Outcome	NOS Total (Max. 9)
Duan et al. [16]	****	*	**	7
George et al. [23]	****		***	7
Hescot et al. [17]	****		*	5
Inguinez-Ariza et al. [18]	**		**	4
Lai et al. [19]	****	**	***	9
Wang et al. [20]	****	**	***	9
Xu et al. [21]	****	**	***	9
Nguyen B et al. [22]	****	**	***	9
Zehir et al. [24]	****	**	***	9

The NOS tool assigns stars (maximum of 9) across three domains: Selection (up to 4 stars), Comparability (up to 2 stars), and Outcome/Exposure (up to 3 stars). Studies scoring ≥7 considered high quality, 4–6 is moderate, and ≤3 is low.

**Table 5 cancers-17-00939-t005:** Summary of findings.

Outcome	Number of Participants (Studies)	Relative Effect (95% CI)	Quality of Evidence (GRADE)	Comments
*BRAF* on OS	877 (6 studies)	1.11 (0.66 to 1.88)	Low	No significant effect, observational studies
*BRAF* on DFS	458 (2 studies)	1.23 (0.66 to 2.29)	Low	No significant effect, observational studies
*TERT* on OS	1023 (9 studies)	1.90 (1.17 to 3.09)	Moderate	Strong association with poorer OS, observational studies
*TERT* on DFS	444 (2 studies)	2.76 (1.86 to 4.10)	Moderate	Strong association with poorer DFS, observational studies
PI3K on OS	661 (6 studies)	2.17 (1.05 to 4.51)	Low	Associated with poorer OS, observational studies
*TP53* OS	686 (7 studies)	2.87 (1.44 to 5.53)	Low	Associated with shorter OS, observational studies

*BRAF*: B-Raf proto-oncogene; PI3K/AKT: phosphatidylinositol-3-kinase/protein kinase B; *TERT*: telomerase reverse transcriptase; *TP53*: tumor protein 53; CI: confidence interval; DFS: disease-free survival; OS: overall survival.

## Data Availability

The data extraction Excel sheets can be provided upon request.

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
