# Peer review of "Prognostic Significance of Key Molecular Markers in Thyroid Cancer: A Systematic Literature Review and Meta-Analysis"

_cancers, 2025, doi:10.3390/cancers17060939_

Round 1

Reviewer 1 Report

Comments and Suggestions for Authors

Nguyen et al, incredibly explored the effect of genetic mutations via NGS on overall survival and disease free survival in thyroid cancer patients . Based on their systematic review analysis TERT, TP53 and PI3K pathways mutations are valuable for risk stratification, while BRAF lacks predictive utility. 

This topic is extremely interesting adding translational knowledge in the agrressive role of these oncogenes . 

The article is well conducted and well written and all the key points are focused.

I have only some minor concerns.

In the oncological setting, clinicians are always searching beneficial pronostic factors to improve survival and to decrease the mortality risk.

Based on the recent papers, it will be nice if the authors add a small paragraph about the possible role on the clonal hematopoeisis on Tp53 alterations.

This point should be improved. 

Author Response

Dear Reviewer,

Thank you very much for your time to review our manuscript and valuable feedback, which is highly appreciated. Your questions or comments are listed below (bold), followed by our response.

In the oncological setting, clinicians are always searching for beneficial prognostic factors to improve survival and to decrease the mortality risk. Based on the recent papers, it will be nice if the authors add a small paragraph about the possible role of clonal hematopoiesis on Tp53 alterations. This point should be improved. 

Thank you for your insightful feedback on our manuscript and for suggesting the inclusion of clonal haematopoiesis (CH) in relation to TP53 mutations in thyroid cancer. I value your perspective and the opportunity to clarify the decision-making process behind the study’s scope. Below, I explain why CH was not included in this paper while recognising its significance and potential for future exploration.

I appreciate your point about the emerging role of CH in oncology. CH is indeed gaining attention as a potential prognostic factor, particularly due to its association with TP53 mutations and its link to increased risk of solid tumours, including thyroid cancer. Research has suggested that CH may contribute to cancer progression through mechanisms like immune dysregulation or changes in the tumour microenvironment. Your suggestion highlights an important consideration, and I understand why CH could enhance the discussion of prognostic factors in this context. After careful deliberation, however, we decided not to mention CH in this meta-analysis due to the following reasons:

Limited Thyroid Cancer-Specific Evidence: Although CH has been associated with solid tumours broadly, the data specifically linking CH and TP53 mutations to thyroid cancer prognosis remain limited. Most research on CH centres on haematological malignancies or general cancer risks, with little direct evidence tailored to thyroid cancer [1]. Incorporating CH would thus require assumptions or extrapolations that might not align with the study’s goal of providing precise, thyroid cancer-specific insights.

While CH is not addressed in this paper, we agree that it holds promise as an area of investigation, particularly for understanding how systemic factors might influence thyroid cancer progression or therapeutic outcomes. We are eager to consider CH in future studies, especially as more data specific to thyroid cancer emerge, and we see it as a valuable direction for expanding this research.

In summary, although CH is a compelling topic, its inclusion would extend beyond the study’s core focus on tumour-specific gene mutations and their prognostic value in thyroid cancer. We believe the current scope maintains a targeted and clinically relevant analysis. Thank you once again for your constructive suggestion—it has enriched our thinking about the study’s boundaries and potential next steps.

[1] Marco M. Buttigieg et al., Clonal Hematopoiesis: Updates and Implications at the Solid Tumor-Immune Interface. JCO Precis Oncol 7, e2300132(2023). DOI:10.1200/PO.23.00132

Reviewer 2 Report

Comments and Suggestions for Authors

The authors present a systematic review and meta-analysis of the prognostic significance of BRAF, TERT, P53 and PI3K in thyroid cancer.

I think the paper is well designed and structured. Introduction and Methods are also clear, Discussion well developed.

However, some istance could be proposed.

First, the decision to include both DTC, PDTC and ATC: what about that? I think DTC and ATC have very different behaviour and prognosis, and including both could make the research rich but perhaps unfocused. 

Secondly, in lines 110-111, why this criterion for DFS? It is certainly not eligible for ATC in its bichemical part. Furthermore, why this cut off for AbTg when considering DTC? To my knowledge there is no AbTg cut off for in the definition of DFS, please provide references or an explanation for this choice (I think it could also be an author's choice, but in that case it needs an explanation.).

Moreover:

  • do you considered articles in language other than english? Please specify.
  • Lines 103-104: what do you mean by "more relevant"? I think it is appropriate to give a short explanation.

I also think that the conclusion, which is clear and supported by results and discussion, could be a little more consistent.

Author Response

Dear Reviewer,

Thank you for your detailed comments and review of our paper. We have found the comments to be of great assistance in strengthening and refining our paper. Your questions or comments are listed below (bold, black), followed by our response and any changes in the manuscript highlighted in YELLOW.

  1. The decision to include both DTC, PDTC and ATC: what about that? I think DTC and ATC have very different behaviour and prognosis, and including both could make the research rich but perhaps unfocused. 

Thank you for your thoughtful and detailed feedback. We acknowledge that DTC, PDTC, and ATC display distinct clinical behaviours and prognostic profiles. DTC is typically indolent with an excellent prognosis, PDTC exhibits intermediate aggression, and ATC is an aggressive, highly lethal malignancy [1,2]. These differences are well-established and critical to consider.

The inclusion of all three subtypes is intentional and aligns with the study’s objectives. Although they differ clinically, these cancers form a spectrum within thyroid cancer, frequently sharing key genetic alterations (e.g., BRAF, RAS, TP53). Including DTC, PDTC, and ATC offers the following advantages:

Comparative Insights: Examining the prognostic impact of shared mutations across subtypes may uncover patterns or markers of disease progression.

Comprehensive Analysis: A broader scope enhances our understanding of mutation significance across the thyroid cancer continuum.

  1. In lines 110-111, why this criterion for DFS? It is certainly not eligible for ATC in its biochemical part. Furthermore, why this cut off for AbTg when considering DTC? To my knowledge there is no AbTg cut off for in the definition of DFS, please provide references or an explanation for this choice (I think it could also be an author's choice, but in that case it needs an explanation.).

Thank you for your comments regarding the definitions of disease-free survival (DFS) for differentiated thyroid cancer (DTC) and anaplastic thyroid cancer (ATC). Below, we provide clarifications to address your concerns and outline the rationale behind these definitions.

Study Context

To begin with, we would like to note that this manuscript presents a systematic review and meta-analysis, rather than a primary research study. As such, the definitions of DFS are crafted to synthesize outcomes reported across a range of studies, accommodating variability in terminology and measurement while ensuring scientific rigor and relevance to each thyroid cancer subtype. We apologize for any ambiguity in the manuscript that may have suggested otherwise, and we will revise the text to explicitly state this focus.

DFS Definition for Differentiated Thyroid Cancer (DTC)

For DTC, we will re-define DFS as follows:

"In studies of differentiated thyroid cancer (DTC), disease-free survival (DFS) was defined as the time from the completion of initial treatment (e.g., surgery, radioactive iodine) to the first occurrence of clinical or radiological recurrence, or biochemical recurrence indicated by elevated thyroglobulin (Tg) levels, with specific Tg thresholds as defined by individual studies."

This definition reflects the standard practice of monitoring DTC recurrence, which includes both structural (clinical or radiological) and biochemical (Tg) indicators. By referencing Tg thresholds as reported in each study, we account for variations in biochemical criteria while aligning with guidelines from the American Thyroid Association (ATA), which recognize Tg as a key marker for DTC recurrence [1,2].

DFS Definition for Anaplastic Thyroid Cancer (ATC)

For ATC, the definition is adjusted to suit its distinct characteristics:

"In studies of anaplastic thyroid cancer (ATC), disease-free survival (DFS) was defined as the time from the completion of initial treatment (e.g., surgery, radiation, chemotherapy) to the first occurrence of clinical or radiological recurrence."

Unlike DTC, ATC does not reliably produce thyroglobulin, making biochemical markers less relevant. Consequently, this definition relies solely on clinical and radiological evidence of recurrence, consistent with clinical follow-up practices for this aggressive subtype [2].

These definitions are intentionally flexible to ensure inclusivity across studies, which may use terms like "recurrence-free survival" or "event-free survival" but report outcomes consistent with our criteria (i.e., clinical, radiological, or biochemical recurrence for DTC; clinical or radiological recurrence for ATC). This approach enables us to integrate a broad evidence base while maintaining clarity and consistency in the meta-analysis.

We hope these explanations clarify the DFS definitions for DTC and ATC and effectively address your concerns. We are eager to revise the manuscript further to enhance transparency or incorporate any additional suggestions you may have. Thank you for your valuable input, which has significantly improved this work.

  1. Do you consider articles in a language other than English? Please specify.

We searched all documents relevant to the research question in the databases, limited to the last 10 years, without excluding any language.

  1. Lines 103-104: what do you mean by "more relevant"? I think it is appropriate to give a short explanation. I also think that the conclusion, which is clear and supported by results and discussion, could be a little more consistent.

Two studies have utilised duplicate data. Wang's study employed MSK MetTropism pan-cancer data, which includes 25775 samples, among which 438 are thyroid cancer samples classified as PTC and PDTC. Meanwhile, Zehir's study used MSK-IMPACT pan-cancer data consisting of 10945 samples, including 231 thyroid cancer samples that encompass both DTC and ATC. All of this data was extracted from cbioportal.com, indicating that the DTC samples might overlap between the two studies.

To address this, we decided to extract the ATC data from Zehir's study while using the PTC and PDTC data from Nguyen's study to maximize the sample size.

  1. I also think that the conclusion, which is clear and supported by results and discussion, could be a little more consistent.

We will make the conclusion more consistent with your suggestion. Thank you.

References

[1] R. Haugen et al., “2015 American Thyroid Association Management Guidelines for Adult Patients with Thyroid Nodules and Differentiated Thyroid Cancer: The American Thyroid Association Guidelines Task Force on Thyroid Nodules and Differentiated Thyroid Cancer,” Thyroid, vol. 26, no. 1, pp. 1–133, Jan. 2016, doi: 10.1089/thy.2015.0020.

[2] K. C. Bible et al., “2021 American Thyroid Association Guidelines for Management of Patients with Anaplastic Thyroid Cancer,” Thyroid, vol. 31, no. 3, pp. 337–386, Mar. 2021, doi: 10.1089/thy.2020.0944.

Reviewer 3 Report

Comments and Suggestions for Authors

Dear Dr Anthony Glover

Thank you for your very informative review regarding an interesting and important issue. Genomic profiling of thyroid cancers is of paramount significance towards efficient therapeutic management and tailored made approach. On the other hand, it is very important that you mention the ATA recommendation against using BRAFV600E alone as a prognostic marker for low-risk thyroid carcinoma due to its low positive predictive value.

Here are some comments

  1. Line 92. Did all the studies use the same gene panel? Moreover did the panels include fusions as well?
  2. Lines 110-111. Please define whether Tg refers to basal or stimulated Tg. Regarding AntiTg did all the studies included use the same method?
  3. Lines 166-167. This should be considered as one of the limitations of the study
  4. Lines 215-217. Was this relationship found in BRAF only cases or this was related to the co-existence of another mutation as well?
  5. Line 225. Was this relationship found in TERT only cases or this was related to the co-existence of  another mutation as well?
  6. Line 235. Was this relationship found in PI3K only cases or this was related to the co-existence of another mutation as well?
  7. Line 242. Was this relationship found in TP53 only cases or this was related to the co-existence of another mutation as well?

Thank you

Author Response

Dear Reviewer,

We would like to express our sincere gratitude for taking the time to review our manuscript. We greatly appreciate your insights and expertise, and we are committed to addressing your valuable feedback thoroughly to improve the quality of our work.

Your questions or comments are listed below (bold, black), followed by our response and any changes in the manuscript highlighted in YELLOW.

  1. Line 92. Did all the studies use the same gene panel? Moreover, did the panels include fusions?

Thank you for your comments regarding the gene panel for each study included in this meta-analysis. Below, we provide clarifications to address your concerns.

The included studies use different gene panels with different numbers of genes. Some of them do not include fusions. However, with the aim of understanding the role of TERT, TP53, and PI3K pathway gene mutations, we only selected studies with panels that contain these genes. Thanks to your insightful feedback, we have revised Table 1 in the manuscript, listing the gene panels used in each study. This will help readers better understand the genetic testing methods employed.

Table 1: Characteristics of nine included studies

Author

Publication year

Country

Cancer

type

N. of patient

Female

Age

Observation

period

Mutation

Outcome

Genes panel

Duan et al.[1]

2019

China

PDTC + ATC

66 (53*)

29

NA

6 years

BRAF, TERT, PI3K, TP53

OS

Nextseq500, 18 genes

Hescot et al.[2]

2022

France

PDTC

40

24

59 (51 - 72)a

57.5 months

BRAF, TERT, PI3K

OS, DFS

Thyroseq V3, 112 genes

Inguinez et al.[3]

2020

US

DTC/PDTC + ATC

55

26

55(14)b

30 years

BRAF, TERT.PI3K

OS

Foundation One 315 genes

Lai et al. [4]

2020

Taiwan

ATC

39 (27*)

20

75 (NA)a

21 years

BRAF, PI3K, RAS, TP53

OS

Targeted NGS, 5 genes, fusion not included

Wang et al.[5]

2022

US

ATC

202

95

66(32-92)a

31 months

BRAF, TERT, PI3K, TP53

 OS, DFS

19 genes, fusion not included. Targeted NGS

Xu et al.[6]

2020

US + AU

ATC

360 (102*)

196

68(29-99)a

9 months

BRAF, TERT, PI3K, TP53

OS

MSK-IMPACT, 468 genes

Nguyen et al.[7]

2022

US

DTC

25775 (307*)

154

NA

38 months

BRAF, TERT, PI3K

OS

MSK-IMPACT, 468 genes

Geoge et al.[8]

2015

US

PTC

256

108

45.3(NA)a

112 months

BRAF, TERT,

OS, DFS

NGS, 46 genes

Zehir et al.[9]

2017

US

ATC

10336 (33*)

14

NA

30 months

BRAF, TERT, PI3K, TP53

OS

MSK-IMPACT, 341 genes

  1. Lines 110-111. Please define whether Tg refers to basal or stimulated Tg. Regarding AntiTg did all the studies included use the same method?

We would like to note that this manuscript presents a systematic review and meta-analysis, rather than a primary research study. As such, the definitions of DFS are crafted to synthesise outcomes reported across a range of studies, accommodating variability in terminology and measurement while ensuring scientific rigor and relevance to each thyroid cancer subtype. We apologize for any ambiguity in the manuscript that may have suggested otherwise, and we will revise the text to explicitly state this focus.

For DTC, we will re-define DFS as follows:

"In studies of differentiated thyroid cancer (DTC), disease-free survival (DFS) was defined as the time from the completion of initial treatment (e.g., surgery, radioactive iodine) to the first occurrence of clinical or radiological recurrence, or biochemical recurrence indicated by elevated thyroglobulin (Tg) levels, with specific Tg thresholds as defined by individual studies."

This definition reflects the standard practice of monitoring DTC recurrence, which includes both structural (clinical or radiological) and biochemical (Tg) indicators. By referencing Tg thresholds as reported in each study, we account for variations in biochemical criteria while aligning with guidelines from the American Thyroid Association (ATA), which recognize Tg as a key marker for DTC recurrence [10], [11].

Meanwhile, we do not have information on method of detecting AntiTg in each paper.

For ATC, the definition is adjusted to suit its distinct characteristics:

"In studies of anaplastic thyroid cancer (ATC), disease-free survival (DFS) was defined as the time from the completion of initial treatment (e.g., surgery, radiation, chemotherapy) to the first occurrence of clinical or radiological recurrence."

Unlike DTC, ATC does not reliably produce thyroglobulin, making biochemical markers less relevant. Consequently, this definition relies solely on clinical and radiological evidence of recurrence, consistent with clinical follow-up practices for this aggressive subtype [10], [11].

Rationale and Flexibility

These definitions are intentionally flexible to ensure inclusivity across studies, which may use terms like "recurrence-free survival" or "event-free survival" but report outcomes consistent with our criteria (i.e., clinical, radiological, or biochemical recurrence for DTC; clinical or radiological recurrence for ATC). This approach enables us to integrate broad evidence base while maintaining clarity and consistency in the meta-analysis.

We hope these explanations clarify the DFS definitions for DTC and ATC and effectively address your concerns. We are eager to revise the manuscript further to enhance transparency or incorporate any additional suggestions you may have. Thank you for your valuable input, which has significantly improved this work.

  1. Lines 166-167. This should be considered as one of the limitations of the study

Yes, you are right. We will include this point in the discussion for the limitation of this study.

“Despite its strengths, this review has several limitations. As shown in the funnel plot in Figure 8, publication bias may have impacted the findings. The included studies exhibited variability in methodology and population demographics, which could affect the generalizability of the results. Furthermore, some mutations, such as those in the PI3K pathway and TP53, were studied in smaller cohorts, potentially reducing the statistical power to detect meaningful associations. Furthermore, the included studies did not provide explicit information on recurrence, which limits our ability to comprehensively evaluate the association between mutations and recurrence patterns, as well as their potential implications.”

  1. Lines 215-217. Was this relationship found in BRAF only cases or this was related to the co-existence of another mutation as well?
  2. Line 225. Was this relationship found in TERT only cases or this was related to the co-existence of  another mutation as well?
  3. Line 235. Was this relationship found in PI3K only cases or this was related to the co-existence of another mutation as well?
  4. Line 242. Was this relationship found in TP53 only cases or this was related to the co-existence of another mutation as well?

We extracted data on the association of each individual mutation with patient overall survival and disease-free survival. This is also a limitation of this meta-analysis when we cannot evaluate the synergic effect of co-mutation on outcomes. This limitation has been stated in the manuscript. Thank you.

References

[1]       H. Duan et al., “Mutational profiling of poorly differentiated and anaplastic thyroid carcinoma by the use of targeted next‐generation sequencing,” Histopathology, vol. 75, no. 6, pp. 890–899, Dec. 2019, doi: 10.1111/his.13942.

[2]       S. Hescot et al., “Prognostic of recurrence and survival in poorly differentiated thyroid cancer,” Endocr Relat Cancer, Aug. 2022, doi: 10.1530/ERC-22-0151.

[3]       N. M. Iñiguez-Ariza et al., “Foundation One Genomic Interrogation of Thyroid Cancers in Patients With Metastatic Disease Requiring Systemic Therapy,” J Clin Endocrinol Metab, vol. 105, no. 7, Jul. 2020, doi: 10.1210/CLINEM/DGAA246.

[4]       W.-A. Lai, C.-Y. Liu, S.-Y. Lin, C.-C. Chen, and J.-F. Hang, “Characterization of Driver Mutations in Anaplastic Thyroid Carcinoma Identifies RAS and PIK3CA Mutations as Negative Survival Predictors,” Cancers (Basel), vol. 12, no. 7, p. 1973, Jul. 2020, doi: 10.3390/cancers12071973.

[5]       J. R. Wang et al., “Impact of Somatic Mutations on Survival Outcomes in Patients With Anaplastic Thyroid Carcinoma,” JCO Precis Oncol, vol. 6, no. 6, Aug. 2022, doi: 10.1200/PO.21.00504.

[6]       B. Xu et al., “Dissecting Anaplastic Thyroid Carcinoma: A Comprehensive Clinical, Histologic, Immunophenotypic, and Molecular Study of 360 Cases,” Thyroid, vol. 30, no. 10, pp. 1505–1517, Oct. 2020, doi: 10.1089/thy.2020.0086.

[7]       B. Nguyen et al., “Genomic characterization of metastatic patterns from prospective clinical sequencing of 25,000 patients,” Cell, vol. 185, no. 3, pp. 563-575.e11, Feb. 2022, doi: 10.1016/J.CELL.2022.01.003.

[8]       J. R. George et al., “Association of TERT Promoter Mutation, But Not BRAF Mutation, With Increased Mortality in PTC,” J Clin Endocrinol Metab, vol. 100, no. 12, pp. E1550–E1559, Dec. 2015, doi: 10.1210/JC.2015-2690.

[9]       A. Zehir et al., “Mutational landscape of metastatic cancer revealed from prospective clinical sequencing of 10,000 patients,” Nature Medicine 2017 23:6, vol. 23, no. 6, pp. 703–713, May 2017, doi: 10.1038/nm.4333.

[10]     B. R. Haugen et al., “2015 American Thyroid Association Management Guidelines for Adult Patients with Thyroid Nodules and Differentiated Thyroid Cancer: The American Thyroid Association Guidelines Task Force on Thyroid Nodules and Differentiated Thyroid Cancer,” Thyroid, vol. 26, no. 1, pp. 1–133, Jan. 2016, doi: 10.1089/thy.2015.0020.

[11]     K. C. Bible et al., “2021 American Thyroid Association Guidelines for Management of Patients with Anaplastic Thyroid Cancer,” Thyroid, vol. 31, no. 3, pp. 337–386, Mar. 2021, doi: 10.1089/thy.2020.0944.

Round 2

Reviewer 2 Report

Comments and Suggestions for Authors

The authors revised the manuscript according to the few suggestions provided and improved the manuscript.

I think they did a good job on an interesting topic and produced a paper with scientific relevance.